# Determination of Runoff Curve Numbers for the Growing Season Based on the Rainfall–Runoff Relationship from Small Watersheds in the Middle Mountainous Area of Romania

Carina Strapazan [1], Ioan-Aurel Irimuș [1,*], Gheorghe Șerban [1,*], Titus Cristian Man [1,*] and Laura Sassebes [2]

1   Faculty of Geography, Babeș-Bolyai University, 5–7 Clinicilor Street, 400006 Cluj-Napoca, Romania
2   Brasov Water Management System, "Olt" Water Basin Administration, Romanian Waters National Administration, 500084 Brasov, Romania
*   Correspondence: aurel.irimus@ubbcluj.ro (I.-A.I.); gheorghe.serban@ubbcluj.ro (G.Ș.); titus.man@ubbcluj.ro (T.C.M.)

**Abstract:** The NRCS-CN (Natural Resources Conservation Service curve number) method, developed by the USDA (U.S. Department of Agriculture) is among the most widely used for the estimation of surface runoff from watersheds. Ever since its introduction in the 1950s, although it has been used to a great extent by engineers and hydrologists, the applicability of the original procedure used to determine its main parameter, the curve number (CN), to various regions with diverse environmental conditions, is still subject to many uncertainties and debates. This study presents a comparative analysis of different methods applied to determine curve numbers from local data in four watersheds located in the central part of Romania, within the mountain region surrounding the Brașov Depression. The CN values were not only computed using rainfall–runoff records from 1991 to 2020, but also determined from the standard NRCS tables documented in the National Engineering Handbook part 630 (NEH-630), for comparison purposes. Thus, a total of 187 rainfall–runoff data records from the study watersheds and five different methods were used to assess the accuracy of various procedures for determining the CN values, namely: tabulated CN (CN values selected from NRCS tables, TAB), asymptotic fitting (AF) of both natural and ordered data, median CN (MD), geometric mean CN (GM) and the arithmetic mean CN (AM) methods. The applicability of the aforementioned methods was investigated both for the original fixed initial abstraction ratio $\lambda = 0.2$ and its adjustment to $\lambda = 0.05$. Relatively similar results were found for the curve number-based runoff estimates related to the field data analysis methods, yet slightly better when the $\lambda$ was reduced to 0.05. A high overall performance in estimating surface runoff was achieved by most CN-based methods, with the exception of the asymptotic fitting of natural data and the tabulated CN method, with the latter yielding the lowest results in the study area.

**Keywords:** initial abstraction ratio; NRCS-CN method; median; asymptotic fitting; geo-mean; Brașov Depression



## 1. Introduction

Over the last two decades, Europe has experienced an increasing frequency and intensity of extreme phenomena such as droughts and heavy precipitation events with associated flash floods and river floods [1]. The increasing potential for flash flood fatalities in many regions [2] has led to the emergence of a large number of studies addressing the applicability of runoff estimation in scarcely gauged, or even ungauged flash flood prone catchments, many of which were conducted in the central [3,4], southern [5] and southeastern regions of Europe [6,7], as well as in the Carpathians [8–10], on the basis of the highly renowned NRCS-CN method.

Given the increased susceptibility to flash flood events in small mountainous watersheds covering areas of less than 200 km$^2$ with many torrential tributaries [11], there is

a crucial need for rainfall–runoff process modelling, especially in Romania, where such catchments are represented by a limited number of monitoring sites.

The NRCS-CN method, formerly known as the SCS-CN, often used for estimating runoff at the watershed scale, developed in 1954 and published by the original Soil Conservation Service in Section 4 of the National Engineering Handbook (NEH-4), relies on a single parameter referred to as the CN, which is derived from the runoff-producing characteristics of watersheds, such as land use, soil type and hydrologic conditions [12,13]. Any changes in the patterns of such variables may significantly influence hydrological processes, since runoff is very sensitive, inter alia, to soil hydraulic conductivity and related properties, the forest coverage rate and vegetation cover [14–16].

Initially developed for application to small-scale agricultural watersheds based on data analysis across the United States [17], over the years, the method's usage has been extended to rural, forested and urban watersheds [18].

Although the applicability of the method and its limitations have long been debated globally [19], in Romania many studies have demonstrated its effectiveness, such as those conducted by Crăciun et al. [20], Domnița et al. [21–23], Strapazan et al. [24–26] and Haidu and Strapazan [27]. However, given that the method development has been mostly based on information collected from agricultural areas, without much emphasis on forested, desert or urban catchments [28], the use of standard NRCS tables for selecting the CN values poses challenges, providing the poorest estimates for forested basins [29]. Although the NRCS tables present a wide range of CN values representative of different land uses, including forested areas, values corresponding to forest cover in tropical and temperate regions have not yet been validated [30]. The NRCS tables originally published in NEH-4 are now given in the NEH-630 documentation [31].

At a local level, establishing the CN parameter based on NRCS tables (tabular CN) involves a number of risks, which is why increasing attention must be paid to the use of this method in different climatic or geographical regions, other than those for which it was developed [17], with calibrations of locally measured data being mandatory [28].

One of the method's main characteristics is the assumption of a constant initial abstractions (Ia) coefficient $\lambda = 0.2$, which formed the basis for the CN tables development [32], the latter being calculated as median values of the measured data [28]. Subsequent studies have demonstrated the shortcomings in applying the method by using this constant value of 20%, corresponding to the ratio of the initial abstractions to the potential maximum retention (S), with superior results being achieved when a lower value of $\lambda = 0.05$ is applied [28,33–37].

In order to analyze the accuracy of tabulated CN values, i.e., to extend the methodology to other regions, there are a number of studies in the literature that have developed or applied different methods to determine the CN values based on measured rainfall–runoff data [17]. Such methods include asymptotic fitting [29,35,37–41], median [19,35,37,39,42,43], geometric mean [19,37,39,43] and arithmetic mean methods [37,39,43]. Most studies have shown that the application of tabulated CN (TAB) values, according to the original NRCS-CN procedure, with $\lambda = 0.2$ is subject to systematic errors in estimating the runoff characteristics of forested catchment areas [19,30,39].

Considering the shortcomings and limitations of the traditional NRCS-CN tabular method (TAB), worldwide application to forested areas, as well as the lack of studies in Romania addressing the calibration and validation of CNs based on measured data, this study aims to evaluate various methods for computing these values based on measured data from four small catchment areas within the eastern Carpathians. The five methods used for the CN estimation are the tabular (TAB), asymptotic fitting of both ordered ($AF_O$) and unordered, natural data series ($AF_N$), median (MD), geometric mean (GM) and the arithmetic mean (AM) methods. This study addresses the applicability of the traditional method of computing CN values based on runoff generation factors (TAB: easily applicable and less time consuming), in comparison to the use of CN values derived from rainfall–runoff records (a cumbersome and time-consuming procedure), from forested, mountainous catchments that have not been studied so far from this perspective.

All the calculations were performed, both for the original NRCS initial abstractions coefficient of λ = 0.2 and for λ = 0.05, often suggested in the literature. A total of 187 *P-Q* (rainfall–runoff) pairs recorded between 1991–2020 were used, based on the availability of data from the stations.

Concerning the analysis of both the ordered and natural, unordered data series, the results revealed a general tendency of CN values stabilization towards higher precipitation amounts and a decreasing trend associated with lower values of rainfall. Within a catchment, the CN is a variable rather than a constant, taking different values from event to event, hence the possibility of a multitude of CNs [17,44].

Methods applied to higher amounts of precipitation produce results corresponding to higher maximum retention values (*S*) [38]. The NEH-4 method (MD-mean of the annual maximums on which the NRCS tables were initially developed) is an approach that excludes the generally known tendency of CN values to decrease with increasing precipitation, which is why it can lead to systematic errors towards higher CN values [28]. These CN values corresponding to significant rainfall events may be reliable for estimating runoff and, therefore, may be of great use in designing hydraulic facilities [45].

Another goal of this paper was to find a representative CN for the largest rainfall–runoff events recorded at the stations within the study area, according to data availability. Thereby, the asymptotic fitting method was applied to all the available 187 *P-Q* events, including those with *p* < 25.4 mm (1 inch), while for the central tendency methods, only the largest 22 (available from the gauge locations in the Teliu and Timiș watersheds) and 17 *P-Q* events (recorded at the hydrometric stations in the Covasna and Ozunca river catchments) were used.

The current study stems from the hypothesis, according to which a relationship exists between the curve numbers and rainfall depth, which can only be defined based on measured data along with the fact that the use of NRCS table values, without site-specific validation, may lead to unreliable results. This work is intended to provide insight into the use and limitations of the NRCS-CN method for runoff estimation in small mountainous watersheds, which would be useful for the future management of water resources in the region.

## 2. Study Area

The present study was carried out in central Romania, on four small-sized watersheds draining the mountain slopes that surround the Brașov Depression. The Ozunca River basin is located within the central group of the eastern Carpathians, while the other three drainage areas originate in the curvature Carpathians (the southern group of the eastern Carpathians). All the studied sites belong to the upper Olt River basin (Figure 1), with small drainage areas, ranging from 36 to 75 km² above the downstream reference gages (computed within a GIS environment).

The mean elevation in the watersheds ranges from 746 m (Ozunca h.b.) to 1108 m (Timis h.b.) a.s.l., with average slopes between 16.66–36.43% (Table 1).

As regards the land cover distribution, the largest portion of the study area is covered by forests (55.5–92.4%), with only small fractions of urban and heterogeneous agricultural land uses (0.5–3.1%—urban areas and 1.9–4.1%—agricultural land). Large areas are covered by pastures within the Teliu basin (20.7%) and by scrub or other herbaceous vegetation associations within the catchments of Ozunca (23.5%) and Covasna (16.5%) rivers, as shown in Table 2. The urban areas are mostly found downstream from the headwaters of each catchment, where the hydrometric stations are located, namely Teliu, Dâmbu Morii, Covasna and Bățanii Mari, which were considered as outlet points.

The selection of the study area was based on the fact that it is represented by a small number of monitoring sites and numerous torrential systems, being prone to flash flood hazards, endangering the local population and activities. Over the last two decades significant rainfall–runoff events have been recorded in 2016, 2018 and 2010 at the gauging stations. Even though each of the watersheds is equipped with a gauging station, their

ungauged, flash flood prone tributaries pose serious threats to the activities conducted by the forestry institutions and the downstream communities within the area.

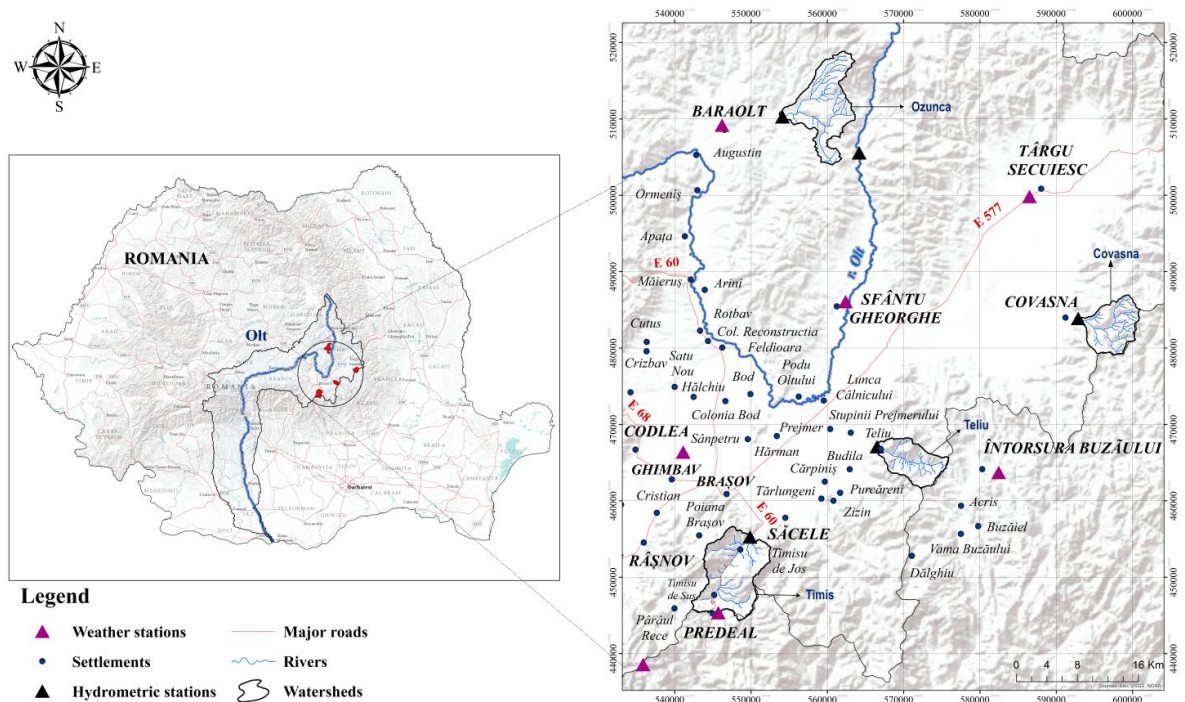

**Figure 1.** General view of the study area and the four watersheds located in the upper Olt River basin.

**Table 1.** Watershed characteristics and data description.

| Watershed | Area (km²) | Mean Elevation (m) | Average Slope (%) | Hydrometric Station | The Time Range of the Observational Data Used in the Study | No. of Events |
|---|---|---|---|---|---|---|
| Teliu | 36 | 801 | 24.92 | Teliu | 1991–2020 | 57 |
| Timis | 75 | 1108 | 36.43 | Db. Morii | 1993–2020 | 64 |
| Ozunca | 66 | 746 | 16.66 | Batanii Mari | 2004–2018 | 34 |
| Covasna | 39 | 1037 | 29.39 | Covasna | 2004–2018 | 32 |

**Table 2.** Land use/land cover distribution.

| Watershed | Forests | | Urban Areas | | Pastures | | Heterogeneous Agricultural Areas | | Scrub and/or Herbaceous Vegetation Associations | | Arable Lands | | Artificial, Non-Agricultural Vegetated Areas | |
|---|---|---|---|---|---|---|---|---|---|---|---|---|---|---|
| | km² | % | km² | % | km² | % | km² | % | km² | % | km² | % | km² | % |
| Teliu | 25.41 | 70.5 | 1.11 | 3.1 | 7.44 | 20.7 | 1.49 | 4.1 | 0.57 | 1.6 | | | | |
| Timiș | 69.31 | 92.4 | 1.90 | 2.5 | 0.60 | 0.8 | 1.39 | 1.9 | 1.21 | 1.6 | | | 0.60 | 0.8 |
| Ozunca | 36.75 | 55.5 | 0.33 | 0.5 | 5.67 | 8.6 | 2.74 | 4.1 | 15.53 | 23.5 | 5.19 | 7.8 | | |
| Covasna | 31.77 | 81.5 | 0.59 | 1.5 | 0.18 | 0.5 | | | 6.43 | 16.5 | | | | |

## 3. Data and Methodology

### 3.1. Data

Given the data availability, the rainfall–runoff events collated for this study case belong principally to the maximum time series length between 1991 and 2020. Daily *P-Q* records compiled from the historical hydrometric archives of the four monitoring sites were used, along with precipitation records from the two closest stations: Micfalău gauging station with influence on the Ozunca catchment and the Predeal weather station located near the

headwaters of the Timiș River. Precipitation spatialization was performed within a GIS environment, using the Thiessen polygon method.

All the data used for the central tendency methods application were subjected to the following analysis procedures:

- historical daily records gathered from April to October were selected, when the watersheds are predominantly rain-dominated;
- only those events with $P > 25.4$ mm (1 inch) [28] and $P/S > 0.46$ [46] were selected, in order to avoid possible bias towards low precipitation amounts. The asymptotic fitting method, used both ordered and unordered datasets for all P-Q pairs;
- partial data pairs were manually removed (e.g., only Q data with missing or inconsistent P data, such as in the case of records from 1993 for the Teliu hydrometric station).

The direct surface runoff was obtained by separating the baseflow from the streamflow hydrograph, using the constant slope model (Figure 2), included in the Cavis software developed by Corbuș [47]:

$$Q_b(t) = Q_b(t - \Delta t) + \Delta Q_{bc} \tag{1}$$

where:

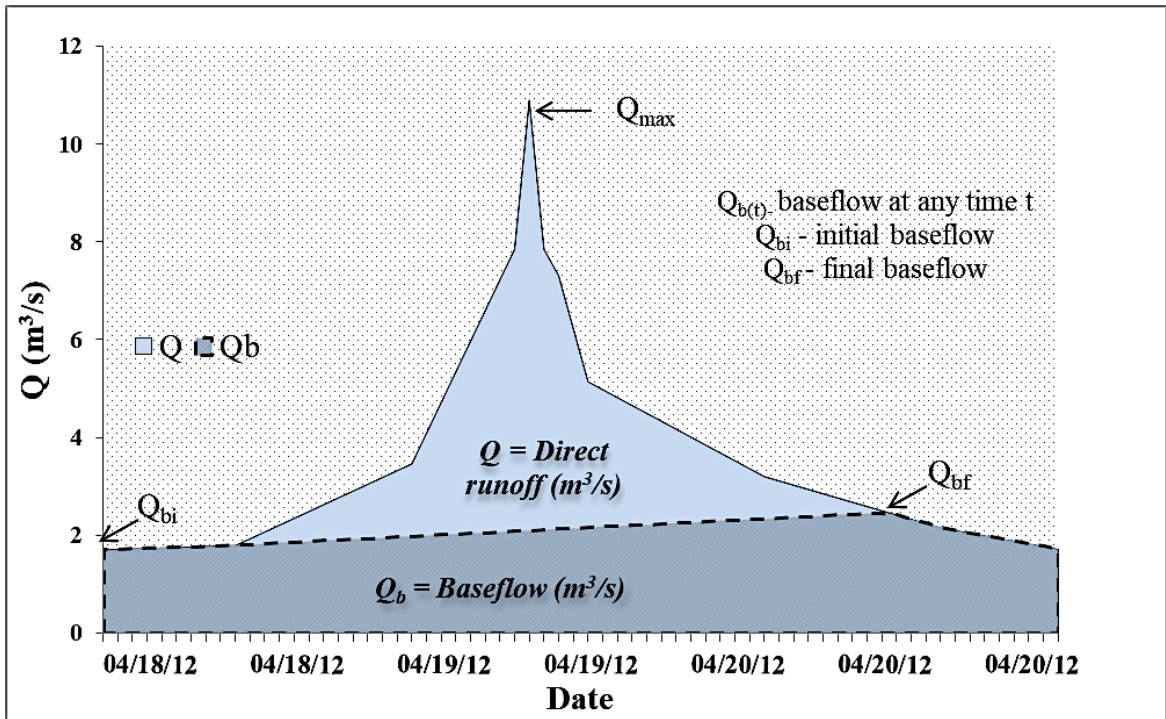

**Figure 2.** Graphical baseflow separation method. An example of the baseflow separation results for the representative runoff event in April 2012 recorded at the Teliu gauge station.

$Q_b(t)$ is the baseflow at any time $t$ (m$^3$/s);
$\Delta t$ is the time step;
$\Delta Q_{bc}$ is the incremental gradient.

Subsequently, 22 events (the largest) were selected from the historical records for the Teliu and Timiș River basins, and 17 events from both the Ozunca and Covasna watersheds, in order to be analyzed using central tendency methods.

One of the purposes underlying the selection of the data for the present analysis was to assess if the CN index varies with the method of determination during significant rainfall events.

### 3.2. NRCS-CN Method

The NRCS-CN method is based on the equation for estimating total runoff from rainfall [31]:

$$Q = \frac{(P - Ia)^2}{P - Ia + S} \; for \; P > Ia,$$

$$Q = 0 \; for \; P \leq Ia,$$

$$Ia = \lambda S \tag{2}$$

$$S = 25.4 \left( \frac{1000}{CN} - 10 \right) \; or \; CN = \frac{25,400}{254 + S} \tag{3}$$

where $P$ is the rainfall amount (mm), $Ia$ stands for the initial losses (mm), $S$ is the potential maximum retention (mm) and $CN$ is the curve number index of the watershed (dimensionless).

The $S$ parameter is a function of land use, soil type, hydrological and antecedent wetness conditions, while $Ia$ refers to the short-term water losses (canopy interception, infiltration), with the NRCS-CN method assuming a constant value of 0.2 for the $\lambda$ coefficient in practical applications [18]. Subsequent evaluations found $\lambda = 0.05$ to be more appropriate, recommending further redefinition of the $CN$ tables [28].

Where data records from the hydrometric and meteorological stations are available, the value of $S$ for $\lambda = 0.2$ can be determined by solving Equation (2) and a sequence of algebraic calculations [29]:

$$S_{0.2} = 5 \left[ P + 2Q - \left( 4Q^2 + 5PQ \right)^{0.5} \right] \tag{4}$$

By substitution, the $CN$ values can be determined directly:

$$CN_{0.2} = \frac{25,400}{254 + 5(P + 2Q - \sqrt{4Q^2 + 5PQ})} \tag{5}$$

In order to adjust the $CN$ and $S$ (mm) parameter values for the assumption of $\lambda = 0.05$ ($CN_{0.05}$ and $S_{0.05}$) directly from the $\lambda = 0.2$ related results ($CN_{0.2}$ and $S_{0.2}$), Woodward et al. [33], further to the results from the analysis on a series of measured $P$-$Q$ data, suggest the following expressions:

$$CN_{0.05} = \frac{100}{1.879(100/CN_{0.2} - 1)^{1.15} + 1} \tag{6}$$

$$S_{0.05} = 0.8187 S_{0.2}{}^{1.15} \tag{7}$$

Considering the median value for the $P$-$Q$ events analysis [33], solving Equation (2) for $\lambda = 0.05$ based on a sequence of algebraic calculations, $CN_{0.05}$ can be determined as follows [19]:

$$CN_{0.05} = \frac{100}{1 + 0.0393701 \left[ 2P + 19Q - (361Q^2 + 80PQ)^{0.5} \right]} \tag{8}$$

### 3.3. CN Determination Methods

3.3.1. Tabulated CN Method (TAB; CN Values Selected from NRCS Tables)

As a major objective, this study sought to assess the accuracy of the traditional procedure of deriving and using CN values from the NRCS tables, published in the current version of NEH-630 (NRCS, 2004), based on local physical geographic features. The NRCS tables have been adapted to the territorial features of Romania by Chendeș [48].

The weighted average CN values for each basin were determined from the analysis and processing of soil maps at a scale of 1:200,000, provided by ICPA Romania (the National Research and Development Institute for Pedology, Agrochemistry and Environmental Protection), and land use at the watershed level, based on CORINE Land Cover (CLC) data from 2006, 2012 and 2018. The aim of using multiple land cover products was to conduct a comparative analysis that could be justified, including through data with similar temporal resolution to that of the *P-Q* series, since the latter were collected over a relatively long-time span. This could translate into corresponding variations in CNs due to CLC revisions, along with the possibility of changes in the land use pattern over time.

Thus, the CN values obtained ($CN_{II}$) using the classical NRCS-CN procedure (CN-TAB) correspond to the so-called normal antecedent moisture conditions ($AMC_{II}$) from earlier versions of the related documentation, the terminology subsequently changing to "antecedent runoff conditions" [28]. The obtained CN values for each watershed, based on different CLC datasets, showed extremely small differences, as follows: 0.2 for Teliu, 0.9 for Timiș, 0.2 for Ozunca and 0.3 for Covasna. Thus, also taking into account the CLC updates and the fact that consistent and high-quality data provided by the Sentinel-2 satellites as part of the European Copernicus Programme was used for the first time in the CLC 2018 project [49], the tabulated CNs presented in this study are those associated with the CLC 2018 land use database (Figure 3).

The standard SCS-CN approach involved converting the CN values from normal ($AMC_{II}$) to dry ($AMC_I$) or wet soil moisture conditions ($AMC_{III}$), based on the 5-day antecedent rainfall depth and the Table 4.2 criteria given in the 1964 edition [50].

Given the subsequent research that highlighted the limited applicability of the suggested antecedent moisture threshold values applicable only to certain small-sized river basins in Texas, along with the existence of many additional precipitation and watershed factors affecting runoff, the table was removed from the later SCS handbook versions [28]. The term "antecedent runoff conditions" came into use in 1993, accounting for the other factors that define the rainfall–runoff relationship [51], the 5-day prior rainfall approach was replaced by a probabilistic interpretation acknowledging all sources of variation [52,53]. However, most related studies and scientific research carried out in Romania has relied on the NRCS-CN tables and the traditional AMC determination method, given the many ungauged or poorly gauged local watersheds and, therefore, the relatively short observational data records. Although the original CN method recommended the use of the 5-day rainfall amount for the AMC, the term antecedent can vary between 5 and 30 days [18], with a significant impact on the results.

For this case study, the effects of the AMC were considered for the application of the standard CN method, through preliminary analysis according to which the *P-Q* events were classified according to both 5- and 10-day rainfall totals. The purpose was to determine whether the increases in surface runoff depth are justified by the AMC classes in relation to precipitation by means of correlation plots for each basin. Figure 4 shows the relationship between runoff and rainfall for both the 5- and 10-day antecedent conditions.

For the Teliu River basin, 62.9–72.3% of the runoff variation is explained by the storm rainfall depth under different AMC, when 10 days of antecedent rainfall are considered (Figure 4a). Under normal antecedent moisture conditions, the correlation between runoff and precipitation is negative.

The coefficient of determination ($R^2$) for the Timiș watershed varies between 0.758 and 0.956, when 10 days of antecedent rainfall are considered, and between 0.761 and 0.844 for the other case.

Considering only the 5 days prior to the onset of runoff, although higher $R^2$ values were found (>0.786) under wet antecedent moisture conditions for almost all the cases considered, the small number of events comes with large uncertainties regarding the reliability of the results. This is well highlighted by the very high $R^2$ values found for the Ozunca River basin, both under normal and wet antecedent moisture conditions for

an extremely small number of events. Higher values for $R^2$ were observed for all the AMC classes associated with 10 days of antecedent rainfall in the case of the Covasna River.

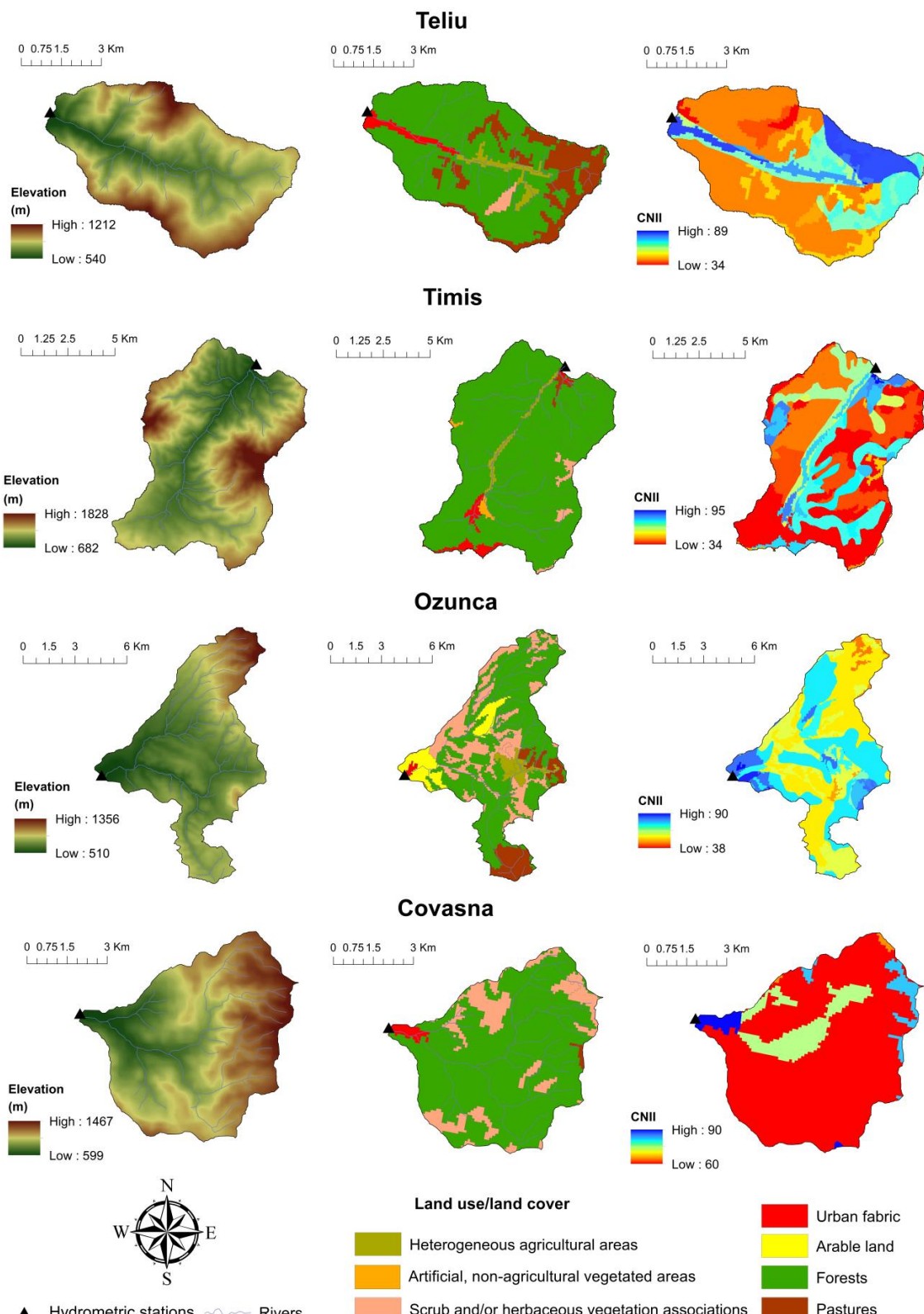

**Figure 3.** Elevation, land use/land cover and CN distribution map for the four selected watersheds.

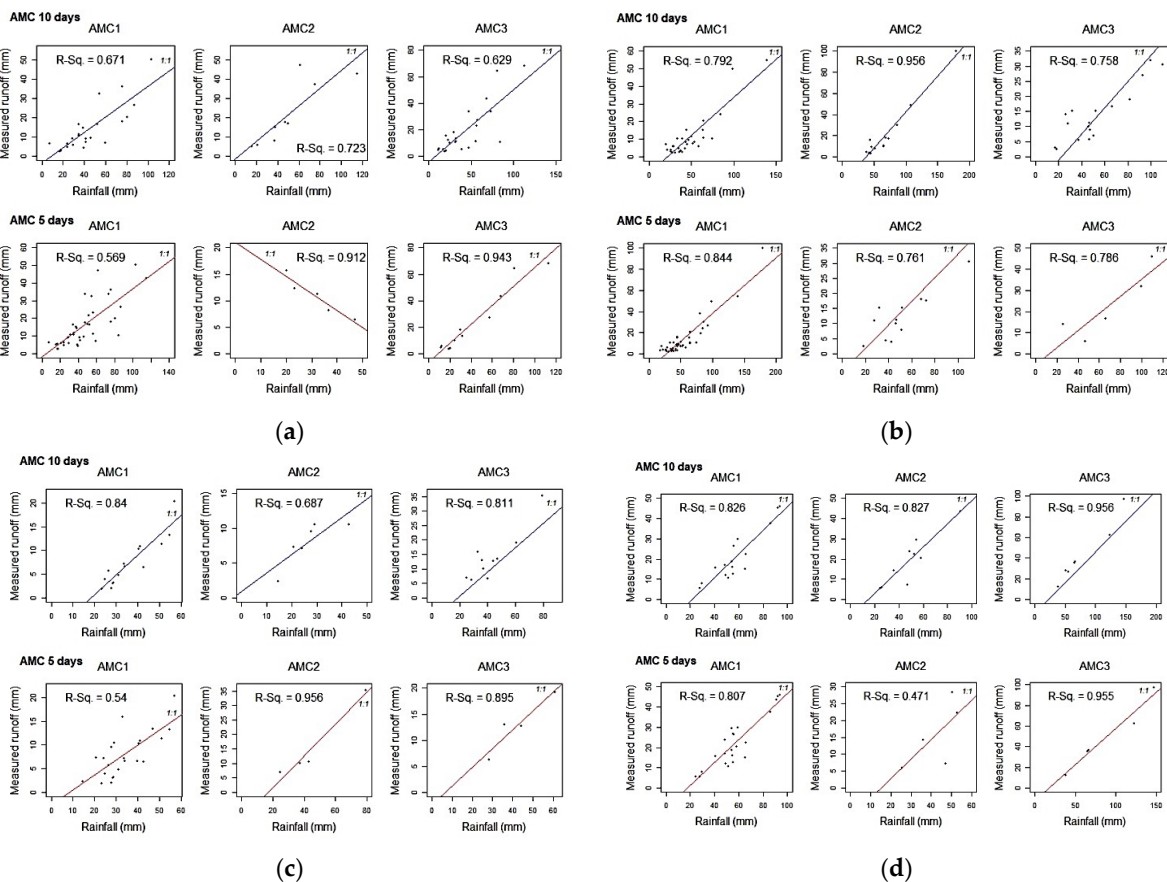

**Figure 4.** Rainfall–runoff relationship among the selected watersheds for the AMC based on the total rainfall, both in the 5-day and in the 10-day period preceding the events: (**a**) Teliu, (**b**) Timiș, (**c**) Ozunca, (**d**) Covasna.

The $CN_{II}$ conversion to dry or wet antecedent conditions was made possible by using the equations recommended by Mishra et al. [19,42]:

$$CN_I = \frac{CN_{II}}{2.2754 - 0.012754CN_{II}} \tag{9}$$

$$CN_{III} = \frac{CN_{II}}{0.430 - 0.0057CN_{II}} \tag{10}$$

### 3.3.2. Median CN Method (MD)

Although the median CN of the observed annual maximum series was the basis for the development of the NRCS tables, the application of the central tendency methods in this study used only the *P-Q* observations from the growing season, as mentioned above.

The S and CN values were determined using Equations (3) and (4), and the CN index was then adjusted for $\lambda = 0.05$ by solving Equation (6). Lastly, the median CN values were extracted for further analysis (CN-MD).

### 3.3.3. Geometric Mean CN Method (GM)

The geometric mean CN was obtained by taking the logarithm of the potential maximum soil moisture retention (*S*) previously determined by means of Equation (4), comput-

ing the arithmetic mean of $log(S)$ and, finally, the geometric mean $S$, $10^{logS}$. Then, the CN (CN-GM) values were determined using the following expression [19,39]:

$$CN_{GM} = \frac{25,400}{\left(254 + 10^{logS}\right)} \tag{11}$$

### 3.3.4. Arithmetic Mean CN Method (AM)

As it is also a central tendency method, the arithmetic mean was used to determine the runoff curve numbers for selected watersheds by first solving the Equations (3) and (4) for $S$ and $CN$, and then extracting the representative CN value from the data series (CN-AM).

### 3.3.5. Asymptotic Fitting Method (AF)

In order to apply this method, all the available 187 events were sorted separately for P and for Q, in descending order, with the CNs found through the use of Equations (3) and (4), corresponding to each re-matched *P-Q* pair of independently ordered data (method AF/CN-$AF_O$). The asymptotic fittings were also carried out using the natural, unordered data series (AF/CN-$AF_N$). The method relies on the assumption that whenever the CN values are determined directly from measured data, a secondary relationship typically emerges between the CN value and the event rainfall depth with the general tendency of CN to approach a constant value asymptotically at higher rainfalls [29]. In other words, the method determines CN as an asymptotic limit as P approaches ∞, with the fitted $CN_\infty$ taken as the final, watershed CN [38].

By plotting the CN against P, Hawkins [29] identified 3 different situations described as complacent, standard and violent.

The complacent situation may occur in certain cases and is characterized by the fact that the found CN does not approach an asymptotic value, probably due to the limited data set which does not yet contain any records of larger storms that would allow for the identification of standard or violent patterns; hence, the alternative approach might be a linear function of the form Q = CP, where C is a coefficient related to the impervious fraction of the watershed [29].

The standard watershed response refers to the situation in which there is a clear tendency for CN to decline with increasing rainfall depth, approaching near-constant values at higher rainfalls, described by Hawkins [29] as follows:

$$CN(P) = CN_\infty + (100 - CN_\infty)\exp(-k_1 P) \tag{12}$$

where $CN_\infty$ is the constant as $P \to \infty$ and $k_1$ is the fitting constant.

The violent response of a watershed is characterized by a sharp increase in the CN values, followed by a tendency to approach a constant value with increasing precipitation depth [29], which is expressed as follows:

$$CN(P) = CN_\infty[1 - \exp(-k_2 P)] \tag{13}$$

where $k_2$ is the fitting constant.

The optimization of the $CN_\infty$ and $k$ values was performed by fitting Equation (12) using the Levenberg–Marquardt algorithm, incorporated within the nlsLM function from the minpack.lm package in the R software [54,55], and the lines of code were executed in RStudio [56].

### 3.4. Statistical Analysis for Performance Evaluation

Regarding the performance assessment of the various methods employed to determine the curve numbers for each watershed, the runoff estimates were compared against the measured data. The assessment on the level of agreement between the estimated and actual runoff observations was caried out using several evaluation metrics, such as the *RMSE* (root mean square error), the *PBIAS* (percent bias), $R^2$ (coefficient of determination),

the *NSE* (the Nash–Sutcliffe model efficiency coefficient) and *d* (the index of agreement), mathematically expressed as follows:

$$RMSE = \sqrt{\frac{1}{N} \sum_{i=1}^{n} \left( Q_i^{obs} - Q_i^{est} \right)^2} \tag{14}$$

$$PBIAS = \left[ \frac{\sum_{i=1}^{n} \left( Q_i^{est} - Q_i^{obs} \right) \times 100}{\sum_{i=1}^{n} Q_i^{obs}} \right] \tag{15}$$

$$NSE = 1 - \left[ \frac{\sum_{i=1}^{n} \left( Q_i^{obs} - Q_i^{est} \right)^2}{\sum_{i=1}^{n} \left( Q_i^{obs} - \overline{Q^{est}} \right)^2} \right] \tag{16}$$

$$R^2 = \left[ \frac{\sum_{i=1}^{n} \left( Q_i^{obs} - \overline{Q^{obs}} \right) \times \left( Q_i^{est} - \overline{Q^{est}} \right)}{\sqrt{\sum_{i=1}^{n} \left( Q_i^{obs} - \overline{Q^{obs}} \right)^2 \sum_{i=1}^{n} \left( Q_i^{est} - \overline{Q^{est}} \right)^2}} \right] \tag{17}$$

$$d = \left[ \frac{\sum_{i=1}^{n} \left( Q_i^{est} - Q_i^{obs} \right)^2}{\sum_{i=1}^{n} \left[ \left| Q_i^{est} - \overline{Q^{obs}} \right| + \left| Q_i^{obs} - \overline{Q^{obs}} \right| \right]} \right] \tag{18}$$

where:

$Q^{obs}$ is the *i*th observed *Q* value;

$Q^{est}$ is the *i*th estimated/calculated *Q* value;

$\overline{Q^{obs}}$ is the mean of the observed series;

$\overline{Q^{est}}$ is the mean of the estimated/calculated series.

The evaluation was based on the following criteria: the lower the RMSE values, the better the model performance; the model fit is considered "unsatisfactory" when PBIAS $\geq \pm 25\%$, "satisfactory" for $15\% \leq$ PBIAS $\leq \pm 25\%$, "good" for $10\% \leq$ PBIAS $< \pm 15\%$ and "very good" when PBIAS $< \pm 10\%$; a model fit is usually considered "satisfactory" when the NSE or $R^2 > 0.50$ [57]. According to the evaluation criteria recommended by Diaz-Ramirez et al. [58], a model is considered to have a poor performance if $0.52 < R^2 < 0.61$, satisfactory if $0.62 < R^2 < 0.72$, good if $0.73 < R^2 < 0.81$ and very good if $0.82 < R^2 < 1.00$, with $R^2 = 1$ indicating a perfect fit to the measured data [19].

Of major importance are also the evaluation criteria proposed by Ritter and Munoz-Carpena [59], used to assess the performance of various CN-based methods in previous studies [19,32,60], according to which a model fit is considered unsatisfactory when NSE < 0.65, acceptable when 0.65 < NSE < 0.80, good for 0.80 < NSE < 0.90 and very good when NSE $\geq$ 90.

The index developed by Willmott [61], *d*, ranges from 0 to 1 and the lower the RMSE value and the closer the *d* index to 1, the more appropriate the CN-based methods in determining the CNs for runoff estimation.

All the calculations were performed using the R software and the hydroGOF (hydrological goodness of fit) package [62].

## 4. Results

For each drainage basin, the CN values were computed using several methods: TAB, MD, GM, AF (AF$_O$ and AF$_N$) and AM. Tables 3 and 4 list the CNs derived from the above-mentioned procedures and the ones determined from the NEH-630 tables for comparative purposes, both for $\lambda = 0.2$ and $\lambda = 0.05$. The obtained results show that CN values range from 50.00 (TAB) to 85.89 (GM) for $\lambda = 0.2$ and from 34.74 (TAB) up to 80.89 (GM) when $\lambda = 0.05$. Hence, the highest CNs were determined by the GM method, whereas the lowest values correspond to the TAB method.

**Table 3.** CN values derived by different methods for λ = 0.2.

| Watershed | MD | GM | AM | AF$_O$ | | AF$_N$ | | Behavior | TAB |
|---|---|---|---|---|---|---|---|---|---|
| | | | | CN$_{AFo}$ (R$^2$, SE) | k (SE) | CN$_{AFn}$ (R$^2$, SE) | k (SE) | | |
| Teliu | 85.85 | 85.89 | 85.06 | 80.45 (0.94, 0.438) | 0.034 (0.002) | 70.00 (0.43, 7.228) | 0.017 (0.007) | Standard | 54.00 |
| Timiș | 76.52 | 79.55 | 77.99 | 71.98 (0.88, 0.442) | 0.038 (0.002) | 68.91 (0.51, 2.081) | 0.029 (0.005) | Standard | 50.00 |
| Ozunca | 83.12 | 84.29 | 83.69 | 79.58 (0.80, 0.664) | 0.049 (0.004) | 73.90 (0.43, 4.970) | 0.030 (0.011) | Standard | 73.00 |
| Covasna | 82.56 | 83.98 | 83.45 | 81.87 (0.23 0.883) | 0.050 (0.011) | 79.77 (0.19 2.311) | 0.034 (0.011) | Standard | 61.00 |

**Table 4.** CN values derived by different methods for λ = 0.05.

| Watershed | MD | GM | AM | AF$_O$ | | AF$_N$ | | Behavior | TAB |
|---|---|---|---|---|---|---|---|---|---|
| | | | | CN$_{AFo}$ (R$^2$, SE) | k (SE) | CN$_{AFn}$ (R$^2$, SE) | k (SE) | | |
| Teliu | 80.88 | 80.89 | 79.81 | 75.61 (0.89, 0.524) | 0.046 (0.003) | 64.18 (0.33 7.504) | 0.022 (0.009) | Standard | 39.03 |
| Timiș | 66.96 | 71.43 | 69.67 | 63.91 (0.64, 0.476) | 0.066 (0.005) | 59.55 (0.30, 2.697) | 0.042 (0.009) | Standard | 34.74 |
| Ozunca | 75.27 | 77.19 | 76.36 | 73.84 (0.29, 0.687) | 0.092 (0.013) | 66.51 (0.24, 5.890) | 0.042 (0.017) | Standard | 62.56 |
| Covasna | 79.04 | 78.97 | 78.23 | 77.09 (0.03, 0.810) | 0.103 (0.051) | 75.26 (0.054, 2.470) | 0.053 (0.025) | Standard | 47.10 |

The CNs estimated by the central tendency methods were generally higher than those computed by the other procedures for both λ thresholds, with the CN-AM values lower than those of CN-GM, but higher compared to CN-MD for two out of the four drainage basins (Timiș and Ozunca).

The CN-TAB values were found to vary substantially between basins according to differences in land use and soil characteristics, over a wide range of 50.00 (Timiș) to 73.00 (Ozunca), with Timiș having the highest percentage of forest cover (92.4%) and Ozunca the lowest (55.5%), since the river drains lower-elevation mountainous areas as compared to the other areas studied.

Forest cover ratios along with the soil hydro-physical properties play a crucial role in the water flow regulation function of watersheds, with significant impacts on the runoff coefficients. In order to assess this impact that natural resources have on the studied watersheds, the 27th of June to 4th of July 2018 reference period was chosen as the study area experienced heavy rainfall due to atmospheric instability with cumulative amounts varying in the 109,3 mm-122.6 mm range over the Timiș, Teliu and Covasna watersheds and a 60.6 mm total over the catchment area of Ozunca. The reference period selection also considered the similarities among the antecedent moisture conditions prior to the onset of each runoff event (events associated with wet antecedent moisture conditions

AMCIII accounting for both the previous 5 and 10 days), according to the traditional approach. Lower CNs and runoff coefficients were found in Timiș, given that the highest forest cover of more than 90% of its watershed's area has developed mainly on sandy loam soils with moderately high infiltration rates. The analysis of the data highlighted, to some extent, a general trend of increased runoff coefficients with decreasing forest cover among catchments, Covasna and Teliu being represented by higher measured CNs, as a result of heavy precipitation with nearly similar totals. The same tendency can be noticed within the complete data series investigated (Table 5). The only exception to this is the Ozunca watershed. Despite it having the highest CN-TAB value, all the measurements and analysis showed lower runoff coefficients for the Ozunca River basin, compared to those of the other areas investigated. A possible explanation might be that the runoff coefficient varies with the slope gradient, not accounted for by the CN method, since the Ozunca catchment covers lower altitudes and gentler slopes (as shown in Table 1), which is more likely to be associated with higher depression storage and smaller runoff coefficients. However, the CN values resulting from the application of all the considered techniques were the closest for Ozunca. Moreover, a large portion of its catchment area is covered by loam and loam to clay loam textured soils falling within HSG (Hydrologic Soil Group) B and C with moderately low to moderately high runoff potential, unlike the Timiș River whose drainage area is mostly dominated by HSG A soils with low runoff potential (65.4%).

**Table 5.** Runoff coefficient and CN values for the selected catchments during the summer 2018 rainfall–runoff event for $\lambda = 0.2$ and $\lambda = 0.05$ and the highest runoff coefficient determined from the entire series of available data.

| Watershed | The Rainfall-Driven Runoff Event that Occurred from 27th of June to 4th of July 2018 | | | | The Entire Rainfall–Runoff Series | |
| --- | --- | --- | --- | --- | --- | --- |
| | Runoff Coefficient (C) | q L/s/kmp | CN 0.2 | CN 0.05 | Highest Runoff Coefficient (C) Value | q L/s/kmp |
| Timiș | 0.42 | 264 | 73.9 | 66.7 | 0.56 | 747 |
| Covasna | 0.51 | 359 | 76.8 | 71.1 | 0.67 | 669 |
| Teliu | 0.61 | 1750 | 83.3 | 79.5 | 0.8 | 1750 |
| Ozunca | 0.32 | 721 | 78.9 | 70.9 | 0.49 | 962 |

Considering the descending sequence of the CN-TAB values, the results are as follows: Ozunca > Covasna > Teliu > Timiș. Unlike the central tendencies used for the analysis of measured data for which the series of CN values revealed the following overall pattern in the obtained results: Teliu > Ozunca > Covasna> Timiș for $\lambda = 0.2$ and Teliu > Covasna > Ozunca> Timiș for $\lambda = 0.05$, with only minor differences in the computed values between Covasna and Ozunca. In fact, no major differences were noticed between the CN values determined by the central tendency methods, comparable results also being found among the basins; however, the values were significantly higher than those of the TAB method.

Just as is the case for the TAB method, all of the data-based CN methods determined the lowest values for the Timiș watershed, owing to its high forest coverage and the associated acid brown, sandy–loam textured soils. By contrast, although the CN-TAB value related to the local environmental characteristics of the Teliu River basin was low, the observed data showed significantly higher values. This could be attributed to meteorological factors, including rainfall intensity, duration and its spatial variations. The curve number variability, moreover, might be associated with the spatio-temporal distribution of the rainfall, the quality of the measured data, as well as the impact of the antecedent precipitation and soil moisture characteristics [17].

In order to determine the behavior pattern of the rainfall–runoff relationship, the CN values were derived both from independently ordered and unordered, natural *P-Q* data pairs. The results revealed that all the watersheds depicted standard behavior. Such

behavioral patterns were observed for λ = 0.2 and λ = 0.05, both in the case of the unordered and ordered data series (Figures 5 and 6), especially for the latter. These findings are consistent with results from other studies conducted in various geographical locations around the world.

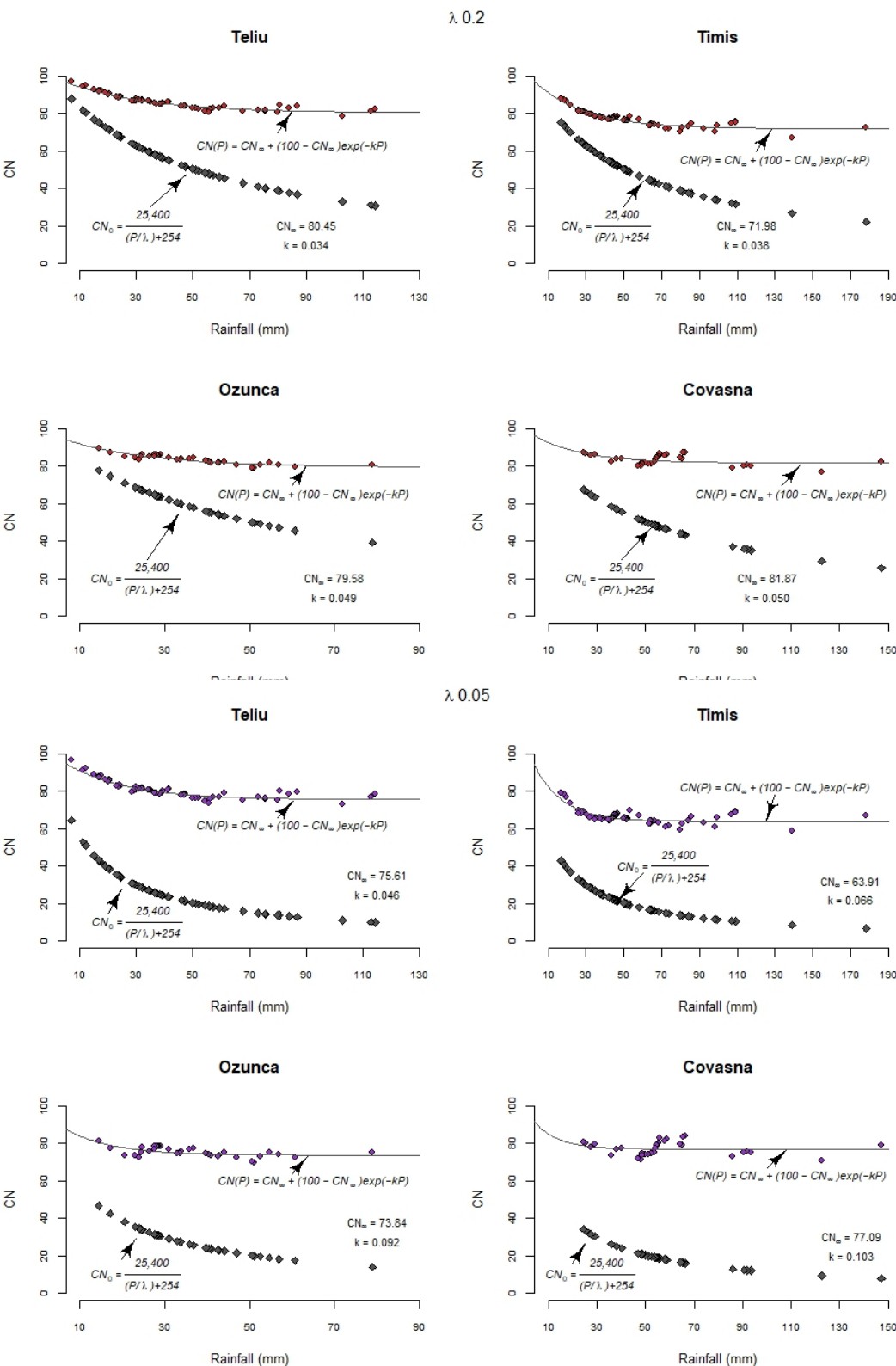

**Figure 5.** CN-AF$_O$, namely the curve number values determined by the asymptotic fitting method for the rank-ordered *P-Q* data pairs, using both values of 0.2 and 0.05 for λ.

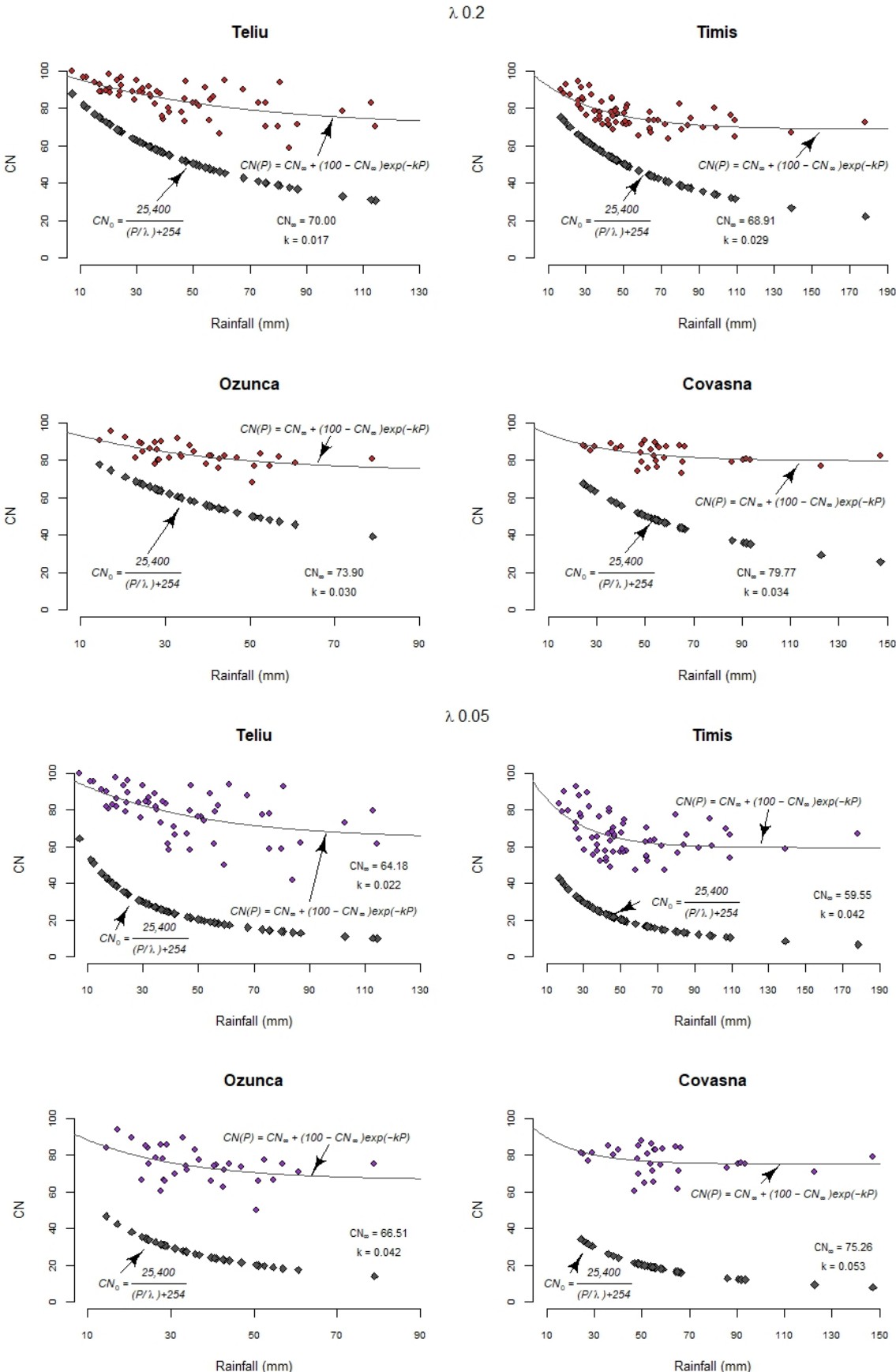

**Figure 6.** CN-AF$_N$, namely the curve number values determined by the asymptotic fitting method for the natural sorting of *P-Q* data pairs, using both values of 0.2 and 0.05 for λ.

Figures 5 and 6 present a greater dispersion of values around the regression equation line of the standard behavior for the natural data sets, compared to the ordered series. Nevertheless, the tendency of CN stabilization at higher rainfall amounts is undeniable.

Considering the CN-$AF_O$ descending sequence for $\lambda = 0.2$ and $\lambda = 0.05$, the following results were achieved: Covasna > Teliu > Ozunca > Timiș, with values that vary between 81.87–71.98 (for $\lambda = 0.2$) and 77.09–63.91 (for $\lambda = 0.05$), while for the CN-$AF_N$, results indicated that Covasna > Ozunca > Teliu > Timiș, varying from 79.77 to 68.91 (for $\lambda = 0.2$) and from 75.26 to 64.18 (for $\lambda = 0.05$). In all cases, lower CNs resulted from the application of the AF method, compared to the central tendency methods. The results showed not only a higher overall scatter corresponding to the $AF_N$ method, but also more spread in the dataset for $\lambda = 0.05$ as opposed to $\lambda = 0.2$. Higher SEs (standard errors of regression) were associated with the $AF_N$ method, especially for $\lambda = 0.05$. $R^2$ (coefficient of determination). The values from Tables 3 and 4 suggest that the standard behavior equation explains better the variance in the CN values for the ordered data sets than for the unordered series.

Regarding the runoff estimation results based on the calculated curve numbers by means of the above-described approaches, the statistical indicators once again highlighted the similarity between the central tendency methods (Tables 6 and 7). Likewise, the boxplot emphasizes the similar distribution pattern for the estimated runoff values based on the CNs derived from the central tendency methods, given the nearly equal interquartile ranges (IQR) and medians (Figure 7). The smallest range of values belongs to the TAB model, with a noticeable smaller height of the box, as compared to those of the other models.

**Table 6.** Accuracy assessment results among the methods used to determine the CN values of the selected watersheds for $\lambda = 0.2$.

| | | $\lambda = 0.2$ | | | | |
|---|---|---|---|---|---|---|
| **Watershed** | **Method** | $R^2$ | **NSE** | **RMSE** | **PBIAS (%)** | $d$ |
| Teliu | MD | 0.751 | 0.678 | 9.850 | 6.6 | 0.924 |
| | GM | 0.751 | 0.677 | 9.870 | 6.8 | 0.924 |
| | AM | 0.751 | 0.698 | 9.550 | 2.2 | 0.927 |
| | $AF_O$ | 0.749 | 0.650 | 10.26 | −20.1 | 0.907 |
| | $AF_N$ | 0.734 | −0.044 | 17.755 | −57.8 | 0.721 |
| | TAB | 0.387 | −0.669 | 22.45 | −69.2 | 0.596 |
| imis | MD | 0.925 | 0.872 | 8.096 | 4.2 | 0.973 |
| | GM | 0.921 | 0.770 | 10.829 | 19.0 | 0.955 |
| | AM | 0.923 | 0.833 | 9.240 | 11.2 | 0.966 |
| | $AF_O$ | 0.929 | 0.890 | 7.487 | −15.5 | 0.974 |
| | $AF_N$ | 0.929 | 0.831 | 9.294 | −27.4 | 0.957 |
| | TAB | 0.313 | −0.122 | 23.948 | −57.5 | 0.635 |
| Ozunca | MD | 0.769 | 0.592 | 4.487 | −0.6 | 0.920 |
| | GM | 0.765 | 0.510 | 4.910 | 8.5 | 0.908 |
| | AM | 0.767 | 0.559 | 4.660 | 3.8 | 0.916 |
| | $AF_O$ | 0.780 | 0.529 | 4.820 | −24.4 | 0.897 |
| | $AF_N$ | 0.796 | −0.128 | 7.458 | −53.5 | 0.751 |
| | TAB | 0.663 | −0.169 | 7.591 | −19.5 | 0.829 |
| Covasna | MD | 0.903 | 0.872 | 7.108 | −3.6 | 0.971 |
| | GM | 0.903 | 0.861 | 7.390 | 2.9 | 0.969 |
| | AM | 0.903 | 0.868 | 7.196 | 0.4 | 0.971 |

**Table 6.** *Cont.*

| Watershed | Method | $R^2$ | NSE | RMSE | PBIAS (%) | d |
|---|---|---|---|---|---|---|
| | | | | $\lambda = 0.2$ | | |
| Covasna | $AF_O$ | 0.903 | 0.867 | 7.230 | −6.7 | 0.969 |
| | $AF_N$ | 0.906 | 0.819 | 8.439 | −15.6 | 0.957 |
| | TAB | 0.784 | −0.329 | 22.877 | −57.2 | 0.779 |

**Table 7.** Accuracy assessment results among the methods used to determine the CN values of the selected watersheds for λ = 0.05.

| Watershed | Method | $R^2$ | NSE | RMSE | PBIAS (%) | d |
|---|---|---|---|---|---|---|
| | | | | $\lambda = 0.05$ | | |
| Teliu | MD | 0.752 | 0.717 | 9.237 | 4.5 | 0.928 |
| | GM | 0.752 | 0.717 | 9.239 | 4.5 | 0.928 |
| | AM | 0.752 | 0.729 | 9.038 | 0.5 | 0.930 |
| | $AF_O$ | 0.745 | 0.394 | 13.529 | −39.7 | 0.830 |
| | $AF_N$ | 0.745 | 0.298 | 14.558 | −43.3 | 0.785 |
| | TAB | 0.590 | −0.715 | 22.755 | −75.9 | 0.612 |
| Timis | MD | 0.925 | 0.916 | 6.537 | −1.7 | 0.980 |
| | GM | 0.921 | 0.893 | 8.414 | 12.9 | 0.970 |
| | AM | 0.923 | 0.911 | 7.396 | 7.0 | 0.976 |
| | $AF_O$ | 0.925 | 0.911 | 6.759 | −10.8 | 0.977 |
| | $AF_N$ | 0.927 | 0.853 | 8.665 | −22.9 | 0.959 |
| | TAB | 0.473 | −0.107 | 23.785 | −65.0 | 0.651 |
| Ozunca | MD | 0.771 | 0.709 | 3.790 | −3.1 | 0.933 |
| | GM | 0.767 | 0.662 | 4.083 | 5.5 | 0.927 |
| | AM | 0.769 | 0.690 | 3.913 | 1.7 | 0.931 |
| | $AF_O$ | 0.773 | 0.708 | 3.792 | −9.1 | 0.930 |
| | $AF_N$ | 0.783 | 0.391 | 5.478 | −34.8 | 0.840 |
| | TAB | 0.678 | 0.085 | 6.716 | −14.3 | 0.853 |
| Covasna | MD | 0.905 | 0.886 | 6.699 | 1.9 | 0.973 |
| | GM | 0.906 | 0.887 | 6.679 | 1.7 | 0.973 |
| | AM | 0.906 | 0.891 | 6.544 | −0.7 | 0.974 |
| | $AF_O$ | 0.906 | 0.891 | 6.556 | −4.3 | 0.974 |
| | $AF_N$ | 0.906 | 0.873 | 7.064 | −9. | 0.968 |
| | TAB | 0.791 | −0.236 | 22.068 | −56.6 | 0.775 |

It should be mentioned that the comparative analysis was carried out on the computed curve numbers previously converted to AMCI or AMCIII on a case-by-case basis, considering the 10-day antecedent rainfall. The reason for this choice is related to the preliminary analysis results presented in chapter 3.

Following the criteria recommended by Moriasi et al. [57], it can be stated that in most cases the models achieved satisfactory results for *NSE* and $R^2$ > 0.50, with the exception of the TAB (for which the results showed less than 0.5 $R^2$ values for two watersheds using λ = 0.2, as well as in the case of λ = 0.05 for one basin and negative *NSE* values in almost all of the cases) and $AF_N$ (with negative *NSE* coefficients found for two of the studied

watersheds using λ = 0.2 and less than 0.5 *NSE* values in two cases for λ = 0.05). Moreover, the TAB and AF$_N$ methods exhibited the highest RMSE and in almost all cases negative PBIAS values that did not fall within the range of ±25%.

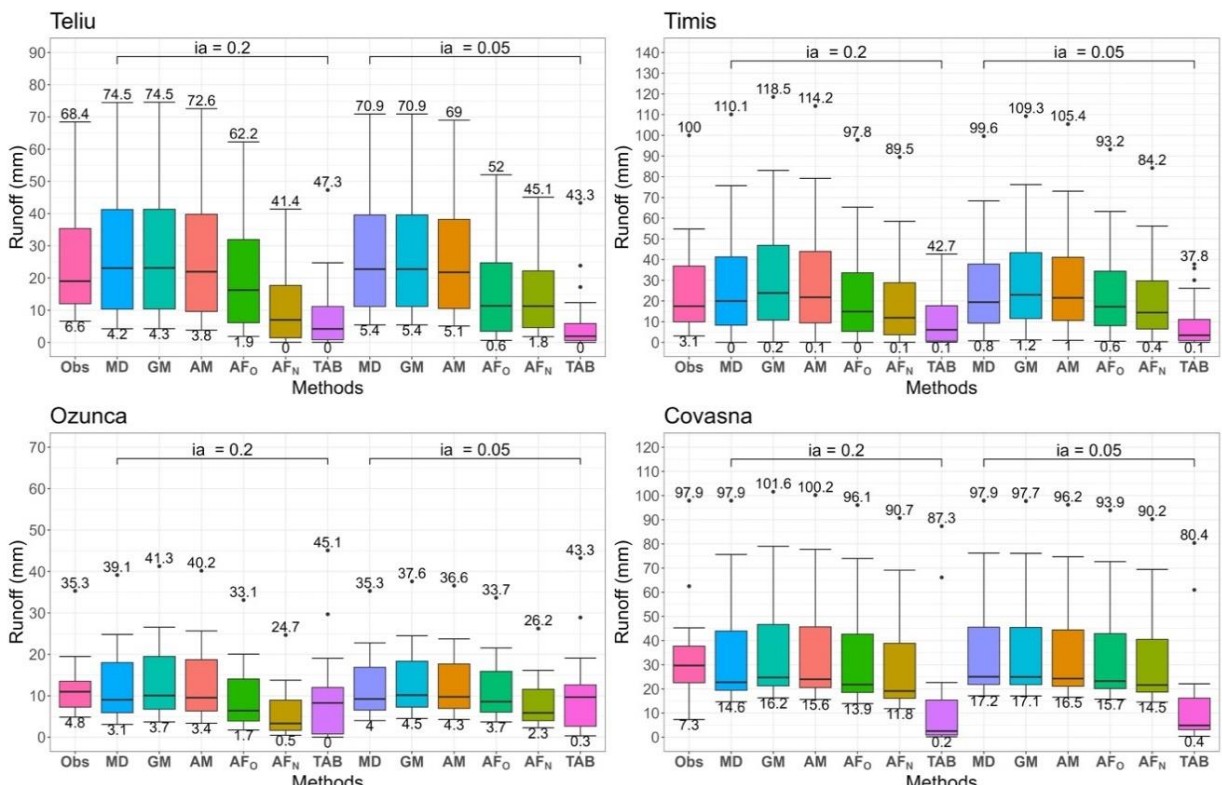

**Figure 7.** Boxplots displaying the distribution of the estimated runoff between the methods employed for the CN determination and the measured runoff (Obs) for λ = 0.2 and λ = 0.05.

Positive PBIAS values indicate model overestimation, while negative values suggest an underestimation [62]. The results showed in most cases a general runoff underestimation associated with the AF$_N$ and the TAB curve number determination methods.

Although high $R^2$ values ($R^2 > 0.5$) were found for Ozunca, Covasna and Teliu (only for λ = 0.05 in this case), the negative *NSE* values indicate substantial biases in the estimations. Consequently, the mean value of the measured runoff data provides a better prediction than the TAB-related estimates compared to the central tendency methods, with no such consistent biases highlighted in the former case by the major dissimilarity between the *NSE* and $R^2$ [39].

Concerning the analysis of the MD, GM and AM methods, the adjustment of the CNs from λ = 0.2 to λ = 0.05 did not make any significant differences in the runoff estimation, $R^2$, *RMSE* and *d*, showing relatively close values. However, slightly larger differences were observed for the *NSE* with higher values at λ = 0.05 and for the *PBIAS* with closer values to the optimal 0. Taking into consideration the criteria suggested by Diaz-Ramirez et al. [58], it may be said that the central tendencies performance is good for Teliu and Ozunca and very good for Timiș and Covasna. Furthermore, following the performance rating suggested by Ritter and Munoz-Carpena [59], the results show higher variability across methods: the performance of the models is acceptable or even good in most of the cases (nine cases) for λ = 0.2, except for Ozunca (*NSE* < 0.65); employing λ = 0.05 leads to acceptable and also good performance in ten cases, and even very good for two cases (Timiș).

Considering the similarity of the results, it is hard to say which of the above-described central tendency methods would be best suited for runoff estimation from higher rainfall amounts in the study area. Yet, better results were obtained using λ = 0.05 with the MD

method for those watersheds larger than 60 km$^2$ (Timiș and Ozunca) and with the AM method for the smaller ones, covering areas of less than 40 km$^2$ (Teliu and Covasna).

Regarding the AF method, significantly better results were achieved for $AF_O$ compared to $AF_N$, as indicated by the higher *NSE* and *d* coefficients and the lower *RMSE* values for all cases.

Although in some cases higher $R^2$ values were given by λ = 0.2, the reduction in the λ value allowed for higher *NSE* and *d* values together with lower *RMSE* and *PBIAS* in all cases. For the most part, the results associated with the $AF_O$ method showed relatively similar $R^2$, *NSE* and *d* to those determined by estimating the runoff based on the central tendencies. The more significant differences lie in the general tendency of the $AF_O$ method-based CNs to underestimate the runoff.

## 5. Discussion

The traditional NRCS-CN tabular method (TAB) exhibited the worst results, since tabular CN estimates provide significantly lower values, as also depicted in other findings [19,30,39,63]. Likewise, Ibrahim et al. [43] also reported the highest CN values as those associated with the GM method for three out of the five investigated watersheds in Sudan and the lowest values corresponding to the tabulated CNs for all cases. Tedela et al. [39] found generally higher CNs for the central tendency methods as compared to the others assessed within the work, among which, the median method provided the lowest values and the geometric mean the highest for four out of the ten studied forested watersheds in the Appalachian Mountains. A general similarity between the CNs derived from the central tendency methods was reported by Im et al. [30], for six experimental forested watersheds located in the temperate climate zone of Japan and South Korea, as stated in the present study. The arithmetic mean and the median of the CN values were also quite close for 10 small-sized Slovak watersheds, but most of the catchments showed complacent behavioral patterns with respect to the asymptotic fitting function, according to the findings of Randusova et al. [63]. However, the standard behavior undeniably seems to be the general pattern, as identified by Ajmal et al. [19] in 80% of the studied watersheds in South Korea, with similar results being reported by Tedela et al. [39] in the United States. D'Asaro et al. [35] determined the standard response in 43 out of 61 Sicilian watersheds (Italy) and Hawkins [29] found the same response in 27 out of the 37 studied watersheds. Similarly, the results of the analysis conducted by Kowalik and Walega [64] on various small watersheds located in southern Poland, revealed a standard response for three out of the four studied. In the assessment of the AF method across five watersheds located in Saudi Arabia, Farran and Elfeki [40] reported higher correlation coefficients and lower *RMSE*s associated with $AF_N$ and λ = 0.2. Regarding the greater dispersion of values around the regression equation for the $AF_N$ method, they state that the underlying reason for such a situation is the CN estimation from the observed natural rainfall and runoff pairs. When the asymptotic fitting was used to evaluate the curve numbers and retention parameters in 10 Slovak and Polish watersheds, Rutkowska et al. [65] reported high $R^2$ values associated with 80% of cases, similar to the present study's findings. In fact, Hawkins et al. [28] state that ordered data series generally provide the most trustworthy and consistent results with higher CN values than those determined by using natural, unordered data. Tedela et al. [39] reported lower CNs resulting from the application of the AF method for all of the studied watersheds, when compared to the central tendency measures. On the other hand, Niyazi et al. [66] state that the asymptotic CN method is not a reliable one for runoff simulation, based on the outcomes exhibited when applied to a Saudi Arabian watershed, but reducing the λ to 0.01 might improve the results in some sense. However, it seems that a reduction in the λ value generally provides better outcomes and the suggested value of 0.05 might suffice. In a research conducted by Baltas et al. [67], aimed at determining the initial abstraction ratio based on rainfall–runoff records from an experimental watershed located in Attica, Greece, λ was found to be very close to 0.05. The fact that the CN adjustment for λ = 0.05 did not reveal significant differences in the runoff estimation performance

related to the central tendency methods was also reported by Ajmal et al. [19], for the South Korean watersheds studied.

*Limitations*

There are no clear indications regarding the applicability limits of the CN method based on watershed size, although a small-scale study may seem like the better option, however it is not always the best one if lumped parameterization is used [28]. Even though an alternative approach for CN estimation is adopted, such as the one based on measured rainfall–runoff data, it is not certain that these particular results can be extrapolated to a generalized larger scale. Taking into account that the studied watersheds drain a quasi-homogeneous lowland mountainous region, the information may be transferred to similar neighboring ungauged catchments, but future work is needed to identify such comparable areas, along with possible ways of extending the applicability of the parameter sets.

At the same time, generalization and extrapolation to a larger scale or watershed involves many uncertainties, since the rivers flow towards the Brașov Depression with significantly different morphological, pedological, land use and climatic characteristics. This low-lying area is subject to intense temperature inversion phenomena, which also reflects the vegetation cover. As air masses cross such a wide area of gently sloping land, they are no longer disturbed by physical barriers like the surrounding mountain range that promotes the development of convective precipitation which often leads to flash flooding. Therefore, the drainage characteristics vary within the region, due to the physiographic transition from mountains to depression and the related environmental heterogeneity. Taken together, these circumstances imply the existence of complex relationships between the factors affecting runoff and, therefore, a variety of CNs. While strong relationships and quite reasonable trends or correlations between variables may be identified for small-sized watersheds, due to their environmental homogeneity, care must be taken when attempting to extrapolate the findings outside their geographical bounds.

It should be remembered that the NRCS-CN method was originally developed based on data gathered from homogeneous catchments in terms of soils and land use in a lumped parameter form [28]. Lumped models cannot represent area heterogeneity and the acceptable lumping amount depends on the scale of the problem, while distributed models account for the spatial variations of the watershed characteristics, but they require more data with higher spatial resolution [68].

By working with coarse resolution data (the lumped approach) at a larger scale within the region, account must be taken of the fact that such procedure will neglect the curve number variability along with other complex local features. However, if a distributed or even semi-distributed runoff modelling approach is assessed, employing high quality, fine-resolution data, trustworthy results may be obtained.

It should be mentioned that this study approached the runoff estimation process from a lumped perspective, since the CN values were determined from point records from the gauges. Applying the methodology to higher gauge density areas, could lead to more trustworthy results, but the reliability of each of the methods should be considered.

As regards the outcomes associated with the TAB method, the relatively coarse spatial resolution of land use and soils data is clearly not the main underlying cause of its poor performance, but higher quality products could bring some improvements. On the other hand, the selection of data for asymptotic fitting procedures is of considerable importance, since various CNs and behavioral patterns can be determined based on the series length, which also plays a major role in the central tendency methods application.

## 6. Conclusions

Although the classical approach of the NRCS-CN method, based on tabulated CN values, is of widespread applicability in various studies, especially in Romania, the present study's results emphasize the need for further research in the field of rainfall–runoff

modelling on small-forested catchments to consider the important use of gauged data with regard to the CN.

The present findings have shown the lower accuracy provided by the traditional procedure of deriving the CN values from the NEH-630 lookup tables employed for direct runoff estimation in the study area, compared to the other methods, both for $\lambda = 0.2$ and $\lambda = 0.05$ (even though a slightly higher correlation has been achieved by reducing $\lambda$). Furthermore, the study conducted by Strapazan et al. [26] on evaluating the performance of several methods, including the SCS-CN one available in the Mike Hydro River–UHM, with regard to surface runoff estimation from the Teliu watershed, showed that the use of the NEH-630-derived CN based on soil and land cover data (CN = 54) led to inefficient outcomes. Extensive calibration efforts have been undertaken to determine the optimal CN value (the obtained CN value of 79 being very close to that estimated in this study by the $AF_O$ method for $\lambda = 0.2$, and not that much smaller than the ones determined by the central tendency methods for Teliu).

Even though the overall comparable accuracies were achieved by the rainfall–runoff data-based CN methods, these were significantly superior to the TAB method, supporting the findings of other studies, such as Ajmal et al. [19] and Tedela et al. [39].

However, among other things, the higher uncertainty associated with estimating the CN from the handbook tables may be linked to the spatial resolution of the input data. Improved runoff estimates may be achieved if more detailed information is provided.

Additional research is recommended regarding the appropriate $\lambda$ values. A reduction in the $\lambda$ value to less than 0.05 may allow for better results from the TAB method. Ajmal et al. [19] demonstrated the applicability of the TAB method for lower $\lambda$ values. There is also the possibility to achieve more varied results by extending the database with more records, or even applying the method to only annual maximum runoff events, indicating the need for future work on this matter.

Despite the overall similarity among the results obtained by the central tendency methods and the comparable runoff estimates derived from the application of the $AF_O$ method, suggesting the relatively minor importance of the choice of method used, there are some small differences given that more accurate results were possible for $\lambda = 0.05$. These differences indicated to some degree the better suitability of the MD method for the larger study watersheds and of the AM method for the smaller-sized ones, respectively.

**Author Contributions:** All authors contributed equally to the preparation of this scientific paper. All authors have read and agreed to the published version of the manuscript.

**Funding:** The publication of this article was supported by the 2023 Development Fund of the UBB.

**Data Availability Statement:** Not applicable.

**Acknowledgments:** The authors would like to express their sincere gratitude to the colleagues from the Olt Basinal Water Administration-Brasov and Covasna Water Management Systems for their cooperation in providing data and information without which this work would not have been possible. Special thanks are also extended to the "Babes-Bolyai University" for the digital technical support provided.

**Conflicts of Interest:** The authors declare no conflict of interest.

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
