# Peer review of "Determination of Runoff Curve Numbers for the Growing Season Based on the Rainfall–Runoff Relationship from Small Watersheds in the Middle Mountainous Area of Romania"

_water, doi:10.3390/w15081452_

Round 1
Reviewer 1 Report
This study is an important contribution to the field of hydrology, as it addresses the issue of accurate surface runoff estimation in areas with limited monitoring and a high risk of flash floods. The NRCS-CN approach is a widely used method for estimating surface runoff from watersheds, but there are still questions and disagreements over the original method used to calculate its key parameter, the curve number (CN), in different places with differing environmental conditions.
The study focuses on deriving CN values from local data in four watersheds located in the mountainous area surrounding the Brasov Depression in the centre of Romania.
The methodology used in the study is rigorous and involves comparing various approaches to derive CN values, including tabulated CN, asymptotic fitting, median CN, geometric mean CN, and arithmetic mean CN. The study evaluates the accuracy of each method by using rainfall-runoff information from 1991 to 2020 and a total of 187 rainfall-runoff data records from the study watersheds. The results show that most of the CN-based methods have good overall performance in estimating surface runoff, except for the asymptotic fitting of natural data and the tabular CN method, which produced the worst results for the study area.
However, while the study's methodology and results are robust, the discussion section could be strengthened. The study does not thoroughly compare its results with other studies in the field, which would provide valuable context and insight into the broader implications of the findings. Therefore a stronger discussion is a must to be incorporated.
Additionally, the study could discuss the limitations of the methods used and potential areas for future research to further improve the accuracy of surface runoff estimation. Hence it is suggested to add limitations section to this manuscript.
Author Response
Dear Reviewers and Editor,
We greatly appreciate the opportunity to revise and resubmit our paper and would like to thank you for taking time in reading it and assist us with constructive comments. We have studied your suggestions carefully and revised the manuscript accordingly. It is our belief that the manuscript is substantially improved as a result of the suggested modifications.
Please find below a point-by-point response (in Blue Font) to each of your concerns (in Black Font).
Again, we would like to express our gratitude for the extremely helpful feedback and hope that the revised manuscript will better suit now the Water journal.
Reviewer #1
However, while the study's methodology and results are robust, the discussion section could be strengthened. The study does not thoroughly compare its results with other studies in the field, which would provide valuable context and insight into the broader implications of the findings. Therefore a stronger discussion is a must to be incorporated.
We would like to apologize for any inconvenience caused and understand the reviewer’s viewpoint here. The results and discussion were initially combined into one section which we fear have ended up causing confusion because of the unorganized information. The main text has been carefully reorganized for clarification as separate sections, that we now hope, allow for greater understanding.
We have also added more references to other papers on this subject:
- Randusova et. al, 2015;
- Rutkowska et al. (2015);
- Niyazi et al. (2022)
- Baltas et al. (2007).
Additionally, the study could discuss the limitations of the methods used and potential areas for future research to further improve the accuracy of surface runoff estimation. Hence it is suggested to add limitations section to this manuscript.
We sincerely thank the reviewer for this helpful suggestion and apologize for failing to offer such valuable details. The limitation section has now been added (Section 4.1)
Reviewer 2 Report
This paper addresses a quite important topic in hydrology of the ungauged watersheds, presents a comparative analysis of 5 different methods applied to determine curve numbers from local data in 4 watersheds lying within the Eastern Carpathians which is represented by a small number of monitoring sites and numerous torrential systems, being prone to flash flood hazard endangering the local society. Manuscript has a good structure and appropriate research design, a high significance of content, and therefore – I recommend it for publishing in the journal Water. However, I have a few comments listed below.
In the Introduction, probably in the first paragraph, the authors should mention this research problem in broader geographical context – Carpathians (recommended: https://doi.org/10.1007/s11069-021-04737-2), Europe (recommended: Mrozik K (2016) Assessment of retention potential changes as an element of suburbanization monitoring on example of an ungauged catchment in PoznaÅ„ metropolitan area (Poland). Rocznik Ochrona Srodowiska 18:188–200) or the South-Eastern Europe (recommended: Chatzichristaki C, Stefanidis S, Stefanidis P, Stathis D (2015) Analysis of the flash flood in Rhodes Island (South Greece) on 22 November 2013. Silva Balcanica 16. 76–86) and then narrow to the mountainous watersheds of Romania. It is necessary to improve table 7 in terms of visibility of data and parameters. I guess that some lines in red are a kind of notice to reviewer or similar and that it is not meant to be published and some references are in red. Please solve this. Finally, the reference list should be prepared according to the instructions for authors.
Author Response
Dear Reviewers and Editor,
We greatly appreciate the opportunity to revise and resubmit our paper and would like to thank you for taking time in reading it and assist us with constructive comments. We have studied your suggestions carefully and revised the manuscript accordingly. It is our belief that the manuscript is substantially improved as a result of the suggested modifications.
Please find below a point-by-point response (in Blue Font) to each of your concerns (in Black Font).
Again, we would like to express our gratitude for the extremely helpful feedback and hope that the revised manuscript will better suit now the Water journal.
Reviewer #2
In the Introduction, probably in the first paragraph, the authors should mention this research problem in broader geographical context – Carpathians (recommended: https://doi.org/10.1007/s11069-021-04737-2), Europe (recommended: Mrozik K (2016) Assessment of retention potential changes as an element of suburbanization monitoring on example of an ungauged catchment in PoznaÅ„ metropolitan area (Poland). Rocznik Ochrona Srodowiska 18:188–200) or the South-Eastern Europe (recommended: Chatzichristaki C, Stefanidis S, Stefanidis P, Stathis D (2015) Analysis of the flash flood in Rhodes Island (South Greece) on 22 November 2013. Silva Balcanica 16. 76–86) and then narrow to the mountainous watersheds of Romania.
We apologize for the lack of information and have now included the references in the first paragraph as suggested. This comment has shown us that our Introduction section was not well substantiated and lacked grounding on the relevant literature. We have therefore expanded our selection of references and updated the Introduction section with more detailed information regarding the increasing vulnerability to flash floods in Europe and the related research on this topic, citing additional sources:
- Hansel et al., 2022;
- Marchi et al., 2010;
- Krvavica and Rubinic, 2020;
- Scopesi et al., 2017;
- Sapountzis and Stathis, 2014;
- Hrabia et al., 2020;
- Vojtek and Vojtekova, 2019.
The introduction section now starts with the European overview of flood and flash flood situation and it continues with the Romanian one in order to enhance the logic flow of the article.
It is necessary to improve table 7 in terms of visibility of data and parameters.
We thank the reviewer for pointing this out. The presented table has now been modified as suggested.
I guess that some lines in red are a kind of notice to reviewer or similar and that it is not meant to be published and some references are in red. Please solve this.
We thank the reviewer very much for catching these confusing errors, which we have now corrected (it was a formatting mistake).
Finally, the reference list should be prepared according to the instructions for authors.
We would like to apologize for our mistake. The reference list has been revised appropriately.
Reviewer 3 Report
Review Report on the Manuscript Number: water-2278555 Title: Determination of Runoff Curve Numbers for the Growing Season based on the Rainfall-Runoff Relationship from Small Watersheds in the Middle Mountainous Area of Romania
The manuscript studies a comparative analysis of different methods applied to determine curve numbers from local data in 4 watersheds lying in the central part of Romania. The topic seems to be appropriate for the journal of Water but there are issues that need to be corrected. In the following, I suggest some possible improvements.
1. Please write more accurate keywords. Like the surface runoff and small watershed, just general keywords which are not appropriate.
2. Line 19-21, These sentences should be moved to the end of the study area section.
3. Line 46-47, It is better to use references that are not used frequently in the text. Please read and add references as follows:
Xiaoming ZHANG,…. (2010) Effects of landuse change on surface runoff and sediment yield at different watershed scales on the Loess Plateau. International Journal of Sediment Research.
Misagh Parhizkar , Mahmood Shabanpour, Manuel Esteban Lucas-Borja , Demetrio Antonio Zema, Siyue Li, Nobuaki Tanaka, Artemio Cerd. Effects of length and application rate of rice straw mulch on surface runoff and soil loss under laboratory simulated rainfall. International Journal of Sediment Research 36 (2021) 468e478.
Admas, M.; Melesse, A.M.; Abate, B.; Tegegne, G. Soil Erosion, Sediment Yield, and Runoff Modeling of the Megech Watershed Using the Geo WEPP Model. Hydrology 2022, 9, 208.
Naser Ahmadi-Sani, Lida Razaghnia and Timo Pukkala, Effect of Land-Use Change on Runoff in Hyrcania. Land 2022, 11(2), 220.
4. Line 126, It is better to write your study hypothesis at the end of this paragraph.
5. Line 128, The selection of the study area was according to what? Field investigation or other researches?
6. Line 333, It is not necessary to present this type of figure (Figure 5) in scientific papers.
7. Line 402 to 406, I think you need to discuss here further.
8. Line 461-466, it would be better if you could compare with the results of other studies.
9. Please explain relation and extension of results obtained from the study to natural conditions in larger scales (scaling).
Author Response
Dear Reviewers and Editor,
We greatly appreciate the opportunity to revise and resubmit our paper and would like to thank you for taking time in reading it and assist us with constructive comments. We have studied your suggestions carefully and revised the manuscript accordingly. It is our belief that the manuscript is substantially improved as a result of the suggested modifications.
Please find below a point-by-point response (in Blue Font) to each of your concerns (in Black Font).
Again, we would like to express our gratitude for the extremely helpful feedback and hope that the revised manuscript will better suit now the Water journal.
Reviewer #3
- Please write more accurate keywords. Like the surface runoff and small watershed, just general keywords which are not appropriate.
We agree and are very grateful to the reviewer for pointing to these inaccuracies. We have therefore replaced the words “surface runoff” with “initial abstraction ratio” (since our study aimed at evaluating, through a comparative analysis, the utility of various CN determination methods, considering different approaches to intial abstraction ratio) and “small watershed” with “median” (due to the fact that it is one of the methods assessed in the paper).
We consider that these chosen keywords better define now the research issue covered by our study.
- Line 19-21, These sentences should be moved to the end of the study area section.
We appreciate the reviewer's suggestion. The sentences have now been moved accordingly.
- Line 46-47, It is better to use references that are not used frequently in the text. Please read and add references as follows:
Xiaoming ZHANG,…. (2010) Effects of landuse change on surface runoff and sediment yield at different watershed scales on the Loess Plateau. International Journal of Sediment Research.
Misagh Parhizkar , Mahmood Shabanpour, Manuel Esteban Lucas-Borja , Demetrio Antonio Zema, Siyue Li, Nobuaki Tanaka, Artemio Cerd. Effects of length and application rate of rice straw mulch on surface runoff and soil loss under laboratory simulated rainfall. International Journal of Sediment Research 36 (2021) 468e478.
Admas, M.; Melesse, A.M.; Abate, B.; Tegegne, G. Soil Erosion, Sediment Yield, and Runoff Modeling of the Megech Watershed Using the Geo WEPP Model. Hydrology 2022, 9, 208.
Naser Ahmadi-Sani, Lida Razaghnia and Timo Pukkala, Effect of Land-Use Change on Runoff in Hyrcania. Land 2022, 11(2), 220.
We agree with the reviewer’s suggestion that other references need to be added. We have therefore changed our selection of references and updated the Introduction section accordingly. References to the works used frequently in the paper, have been removed and the suggested ones included . Please also note that one more reference on the topic was added: Singh et al., 2021, in view of the fact that it provides a detailed description of the historical background and fundamental concept underlying the NRCS-CN method referred to in paragraph 3.
- Line 126, It is better to write your study hypothesis at the end of this paragraph.
We agree with the reviewer on this important point and have formulated the research hypothesis at the end of the Introduction section, to address this issue.
- Line 128, The selection of the study area was according to what? Field investigation or other researches?
We understand the importance of clarifying the main purpose behind the selection of the study area and provided further justifications at the end of the Study Area section. We apologize for not making it clear from the beginning. The most important factor in the selection of the study area is the presence of many torrential systems endangering the local communities and forestry activities. Moreover, significant runoff events have been recorded over the past few decades, posing a potential increased risk in the future.These details have now been included in the text.
- Line 333, It is not necessary to present this type of figure (Figure 5) in scientific papers.
We appreciate and we thank the reviewer for the suggestion. The figure has been removed from the paper, and the other ones renumbered accordingly. We are grateful also because this suggestion made us realize that we missed to give credit to the R software used in the paper. Accordingly, 2 new references have been included in the text: R Core Team, 2022 and R Studio Team, 2022.
- Line 402 to 406, I think you need to discuss here further.
We agree with the reviewer’s suggestion and have elaborated on the matter by adding an explanation accordingly.
- Line 461-466, it would be better if you could compare with the results of other studies.
We thank the reviewer for his valuable suggestion. Unfortunately not many works addressed the quality of the asymptotic fit itself by comparing the standard errors of regression and R2 associated with the fitting equations of both natural and ordered data. Most of the works have only focused on comparative analyses of the computed curve numbers by various procedures in relation to their capability to accurately estimate surface runoff. This is justified by the fact that the best-fit CN values either from ordered or natural series may not necessarily be the best CN for use in runoff estimation models (which is also shown by our results, given that a better suitability of the central tendency methods in runoff estimation was noted). Yet, we found it necessary also to pay heed to the quality of the regression model, since the AFO and AFN methods exhibited comparatively different values.
Nevertheless, 2 citations on relevant studies conducting such analyses were added: Farran and Elfeki (2019-which was already cited once but details added on this matter) and Rutkowska et al. (2015). Please note that these citations are now part of the Discussion section, as the manuscript was reorganized as per Reviewer#1 suggested:
“Regarding the greater dispersion of values around the regression equation for the AFN method they state that the underlying reason for such a situation is the CN estimation from the observed naturally rainfall and runoff pairs. When the asymptotic fitting was used to
evaluate curve numbers and retention parameters in 10 Slovak and Polish watersheds, Rutkowska et al. (2015) reported high R2 values associated with 80% cases, similar to the present study's findings.”
- Please explain relation and extension of results obtained from the study to natural conditions in larger scales (scaling).
We sincerely thank the reviewer for this valuable suggestion and apologize for failing to mention such important details. The main text has been carefully reorganized for clarification, as per Reviewer#1 suggested, so that the relation and extension of our results to natural conditions in larger scales has now been addressed in Section 4 .1 (Limitations).
Round 2
Reviewer 1 Report
Thank you for considering my comments on the earlier version of this manuscript